# Valorization of Pineapple Leaves Waste for the Production of Bioethanol

**DOI:** 10.3390/bioengineering9100557

**Published:** 2022-10-15

**Authors:** Reetu Saini, Chiu-Wen Chen, Anil Kumar Patel, Jitendra Kumar Saini, Cheng-Di Dong, Reeta Rani Singhania

**Affiliations:** 1Institute of Aquatic Science and Technology, National Kaohsiung University of Science and Technology, Kaohsiung City 81157, Taiwan; 2Sustainable Environment Research Center, National Kaohsiung University of Science and Technology, Kaohsiung City 81157, Taiwan; 3Department of Marine Environmental Engineering, National Kaohsiung University of Science and Technology, Kaohsiung City 81157, Taiwan; 4Centre for Energy and Environmental Sustainability, Lucknow 226029, India; 5Department of Microbiology, Central University of Haryana, Mahendergarh 123031, India

**Keywords:** pineapple leaves waste, bioethanol, Saccharomyces, separate hydrolysis and fermentation, enzymatic hydrolysis, hydrothermal pretreatment

## Abstract

Being a lignocellulose-rich biomass, pineapple leaves waste (PL) could be a potential raw material for the production of biofuel, biochemicals, and other value-added products. The main aim of this study was to investigate the potential of pineapple leaves in the sustainable production of bioethanol via stepwise saccharification and fermentation. For this purpose, PL was subjected to hydrothermal pretreatment in a high-pressure reactor at 150 °C for 20 min without any catalyst, resulting in a maximum reducing sugar yield of 38.1 g/L in the liquid fraction after solid-liquid separation of the pretreated hydrolysate. Inhibitors (phenolics, furans) and oligomers production were also monitored during the pretreatment in the liquid fraction of pretreated PL. Enzymatic hydrolysis (EH) of both pretreated biomass slurry and cellulose-rich solid fraction maintained at a solid loading (dry basis) of 5% wt. was performed at 50 °C and 150 rpm using commercial cellulase at an enzyme dose of 10 FPU/gds. EH resulted in a glucose yield of 13.7 and 18.4 g/L from pretreated slurry and solid fractions, respectively. Fermentation of the sugar syrup obtained by EH of pretreated slurry and the solid fraction was performed at 30 °C for 72 h using Saccharomyces cerevisiae WLP300, resulting in significant ethanol production with more than 91% fermentation efficiency. This study reveals the potential of pineapple leaves waste for biorefinery application, and the role of inhibitors in the overall efficiency of the process when using whole biomass slurry as a substrate.

## 1. Introduction

Taiwan is the third-largest producer of pineapple, with an annual pineapple production of nearly **42,000** tons in 2019 [1,2]. Taiwanese pineapples have a delicate texture and balanced flavor that make them among the world’s best. Pineapple fields are primarily located in the central and southern regions of the country [1]. The cultivation of pineapple generates a large amount of plant waste. Approximately 250 metric tons of wet plant residues per hectare are generated, which mainly comprise leaves. The crown of the pineapple itself accounts for 10–25% of the total weight of the pineapple [2] The pineapple’s crown is made up of two types of leaves which vary in their length. Long leaves are utilized by many companies for the production of fibers for various applications; in contrast, the short leaves are not suitable for fiber production and are, therefore, left behind causing a disposal problem. These leaves contain a significant amount of cellulose and hemicellulose that can be converted enzymatically into fermentable sugars for the production of value added-products, such as bioethanol [3]. Thus, bioconversion of pineapple **leaves** (PL) waste into bioethanol could help resolve the issue of waste disposal as well as adding value to improve the bioeconomy. It will also help to mitigate the environmental and socio-economic challenges arising due to heavy dependence upon fossil-based fuel resources.

The release of greenhouse gases (GHG) during combustion of fossil fuels is a major cause of global warming and growing environmental concerns which has forced researchers to look for alternative and/or supplementary sources of fuels and energy [4,5]. The second-generation bioethanol is one such potential alternative that can reduce the dependency on petroleum-based products and mitigate the emission of greenhouse gases [6] The second-generation bioethanol is derived from lignocellulosic biomass such as crop residues, woody crops, or energy grasses and is gaining momentum as lignocellulosic biomass is the most abundant and ubiquitous raw material available for mankind [4,7]. The production of bioethanol depends on the availability of the feedstock and conversion technology [8]. Several economic and technological challenges such as pre-treatment, enzyme production, hydrolysis process, fermentation, and distillation present challenges which need head-to-head resolution [9]. During bioethanol production, pre-treatment is the first and essential step because of the recalcitrant structure of the lignocellulosic biomass. Various pretreatment methods are currently in use to deconstruct the tough structure of lignocellulosic biomass for easy access to enzymes [10,11]. The three main components of lignocellulose are: cellulose, which consists of anhydrous glucose units that are linked together with β-1–4 linkage; hemicellulose, which is a heteropolymer that contains both hexose and pentose sugars; and lignin, which is a highly cross-linked polyphenolic compound. Trans-coniferyl, trans-sinapyl, and trans-p-coumaryl alcohols are the main components of lignin, having an important role in the build-up of the structure of plant cells [12]. Cellulose and hemicellulose are tightly bound with lignin and make a tough structure for plant cell walls that is not easy to deconstruct, and needs pretreatment prior to enzymatic hydrolysis [13]. Many physicochemical pretreatment methods, such as dilute acid or alkali pretreatments, are popular due to their technical and economical ease. However, physical methods have the advantages of being environmentally friendly and corrosion free. Hydrothermal pretreatment is one such physical method for the deconstruction of biomass in which hydronium ions are generated during protonation of water when treated at a high temperature in a reactor [14]. In this process, water at subcritical temperature acts as an acid and results in significant changes in the lignocellulosic biomass, thereby enhancing subsequent enzymatic hydrolysis. Many previous reports on the use of hydrothermal pretreatment of lignocellulosic wastes have employed externally added catalysts such as acids or bases [15,16], which makes such studies less environmentally friendly.

PL biomass is available in abundant supply in Taiwan and is a potential lignocellulosic resource for cellulosic bioethanol production. Therefore, the main aim of this study was to valorize PL residues for the production of bioethanol after subjecting it to hydrothermal pretreatment without any externally added acid or base catalyst. The feasibility of enzymatic conversion of both the pretreatment slurry and the solid biomass after pretreatment was evaluated for efficient bioethanol production. Thus, the novelty of this study includes hydrothermal pretreatment of PL without any external acid or base catalyst as well as depicting the differences in hydrolysis of the solid fraction and slurry based on inhibitors present. The presence of inhibitors in the slurry affected the hydrolysis efficiency to an extent; however, no significant effect on fermentation efficiency was observed.

## 2. Materials and Methods

### 2.1. Materials

The ground pineapple **leaves** waste used in this study was received as a gift from TemTec Co. Ltd., Taipei, Taiwan. All medium components and reagents used in this study were of analytical grade. Oligosaccharide standards (arabinotriose, arabinotetrose, arabinotetraose, arabinopentose, mannonbiose, mannontriose, mannontetrose, cellobiose, and cellotetrose) were of HPLC grade and purchased from Biosynth^®^. Inhibitors (furfural, acetic acid, and formic acid) and antibiotics (amoxicillin and streptomycin) were purchased from Sigma-Aldrich, St. Louis, MO, USA. The commercial cellulase of ***Trichoderma reseei*** ATCC-26921 from Sigma-Aldrich (USA), having cellulase enzyme activity of 125 FPU/mL (measured according to the Ghose method) [17], was used for the hydrolysis experiment.

### 2.2. Microorganism and Culture Conditions

*Saccharomyces cerevisiae* strain WLP300 (White Labs Brewing Co., Germany), a kind gift from Prof. Chien-Hui Wu, Department of Seafood Science, NKUST, Kaohsiung, Taiwan, was used in this study for ethanol fermentation of the sugar-rich enzymatic hydrolysate of pretreated PL. The yeast was maintained in yeast-extract peptone dextrose (YPD) medium having (g/L) yeast extract 10, peptone 20 and dextrose 20. The fermentation media contained yeast extract and peptone at 10 and 20 g/L, respectively.

For inoculum preparation, yeast cells were added to 4 × 250 mL Erlenmeyer flasks filled with 100 mL YPD medium, and incubated at 30 °C for 24 h. After incubation, the cell pellet was aseptically collected by centrifugation of the yeast culture at 10,000× *g* and 4 °C in sterile tubes. The pellet was resuspended in 0.9 % (*w/v*) sterile saline to obtain the cell suspension for use as inoculum.

### 2.3. Hydrothermal Pretreatment of Pineapple Leaves Waste Biomass

Ground pineapple **leaves** biomass waste was pretreated by using a 1:10 *w/v* (dry biomass) mass ratio of biomass to water in a 500 mL micro autoclave reactor (ZZKD Instrument Equipment, Zhengzhou, China) with a working volume of 250 mL at 150 °C for 20 min [18]. After the pretreatment, the solid residues were separated from the liquor by centrifugation (8000× *g* for 15 min at 4 °C). The recovered solids (83 g per 100 g of raw biomass) were washed exhaustively with water until neutral pH and squeezed before preservation at −20 °C for further use. Chemical composition (glucan, xylan and arabinan) of the pretreated and raw or native PL biomass was determined (see section on analytical methods). Quantification of sugars, oligosaccharides and inhibitors generated during pretreatment was carried out by High Performance Liquid Chromatography (HPLC) analysis of **the autohydrolysis** pretreatment **liquor** (obtained after the separation of solids from pretreatment slurry) after filtration through 0.22 μm nylon syringe filters. 

### 2.4. Enzymatic Hydrolysis of Pretreated Biomass

Enzymatic hydrolysis of the pretreated **slurry** (containing both solid and liquor obtained as such after pretreatment) as well as cellulose rich solid fraction were carried out in 250 mL Erlenmeyer flasks with a working volume of 100 mL, solid loading of 5% dry wt., commercial cellulase enzyme dose of 10 FPU/gds and pH maintained at 4.8 (with 0.05 M citrate buffer). The flasks were incubated in an orbital shaker maintained at 50 °C and agitation speed of 180 rpm. Aliquots of the enzymatic hydrolysates were withdrawn at 0, 6, 12, 24, 48, and 72 h, filtered using a 0.22-micron syringe filter, subjected to heat treatment at 100 °C for 2–3 min for enzyme inactivation and stored at −20 °C for further analysis. For quantification of the sugars (pentoses, hexoses and oligosaccharides), enzymatic hydrolysates were subjected to HPLC analysis. Total reducing sugars were estimated by the dinitrosalicylic acid method using glucose as standard Miller [19].

### 2.5. Separate Hydrolysis and Fermentation

Fermentation of the enzymatic hydrolysate of the PL biomass was performed by separate hydrolysis and fermentation of pretreated PL biomass with ***Saccharomyces cerevisiae*** strain WLP300. Prior to fermentation, the hydrolysate was supplemented with nutrients and inoculated with 10% (*v/v*) inoculum of the yeast. Incubation was performed at 30 °C for 72 h with agitation speed of 150 rpm. The samples were withdrawn at different time intervals (0, 24, 48, and 72 h) for the quantification of sugars and ethanol by HPLC analysis.

### 2.6. Analytical Methods

#### 2.6.1. Chemical Characterization of Biomass

The moisture content of the biomass was determined by drying it overnight at 60 °C in a hot air oven until constant weight. Compositional analysis of the biomass was carried out to determine the cellulose, hemicellulose and lignin contents of untreated and hydrothermal pretreated PL biomass as per standard protocols [20].

#### 2.6.2. HPLC Analysis

Quantitative estimation of sugars (pentoses, hexoses and oligomers) and inhibitors (acetic acid, furfural and formic acid) in the liquor obtained after separation of solid from pretreatment slurry as well as sugars and ethanol produced during enzymatic hydrolysis and fermentation, respectively, was performed on Agilent Technology 1260 infinity series HPLC system equipped with a refractive index detector and Coregel 87H3 column maintained at 60 °C. Samples were filtered through a membrane filter of 0.45 μm before injection and were eluted with 0.005 M sulfuric acid as mobile phase at a flow rate of 0.65 mL/min. Calibration curves were prepared by using respective standards of sugars, oligomers, ethanol and inhibitors. 

#### 2.6.3. Calculations

The saccharification yield (cellulose to glucose), ethanol yield, productivity and fermentation efficiencies during hydrolysis and ethanol fermentation were calculated by using Equations (1)–(4), as follows:Saccharification yield (%) = [experimental sugar yield, g/L]/[theoretical sugar yield, g/L] × 0.9 × 100(1)
Ethanol yield = [experimental ethanol titer, g/L]/[theoretical ethanol yield, g/L](2)
Ethanol productivity (g/L/h) = [ethanol titer, g/L]/[fermentation time, h](3)
Fermentation efficiency (%) = [ethanol titer, g/L]/[(Initial biomass, g/L) × f × 1.111 × (0.511)] × 100(4)
where f is the cellulose fraction of dry biomass, 0.511 is the conversion factor for glucose to ethanol based on stoichiometric biochemistry of yeast and 1.111 is the conversion factor for cellulose to equivalent glucose. 

## 3. Results and Discussion

### 3.1. Hydrothermal Pretreatment of PL Waste and Biomass Composition 

The compositional analysis of untreated pineapple **leaves** biomass determined on a dry weight basis revealed glucan content of 56.90 ± 2.10 %, xylan 10.88 ± 0.35 % and lignin 14.2 ± 0.42%. Slight variations in the compositional analysis of pretreated pineapple biomass from earlier reports [21,22] could be possibly due to different agro-climatic conditions during pineapple cultivation and the variety of the pineapple used.

Since, the native lignocellulosic biomass is recalcitrant to enzymatic hydrolysis and subsequent bioconversion to ethanol, the PL waste biomass was subjected to hydrothermal pretreatment at 150 °C for 20 min in this study based on the previously used similar conditions for hydrothermal pretreatment of various lignocellulosic biomass for high recovery of reducing sugars. For example, Pinheiro et al. [18] reported the use of similar conditions for the hydrothermal pretreatment of brewer’s spent grain and Imman et al. [16] pretreated the pineapple leaves at 120–160 °C for 20–50 min. (However, the present study did not employ any acid or base catalyst during hydrothermal pretreatment as has been done previously by [16] and others.) Similar conditions were selected to hydrothermally treat the pineapple leaves waste in this study. Following solid-liquid separation after pretreatment, the liquid phase was characterized by HPLC analysis for the estimation of released inhibitors, sugars and oligomers. HPLC analysis revealed glucose, xylose and arabinose contents of 15.95, 20.7 and 2.0 g/L, respectively, in the pretreatment hydrolysate. Low concentrations of a few inhibitors and oligomers were also observed. Specifically, arabinotriose, xylotriose and cellobiose were present at concentrations of 2.0, 59.0 and 287.0 mg/L, respectively, while other oligomers were not detected. Similarly, formic acid and furfural were observed at concentrations of 13.0 and 238.0 mg/L, respectively. It was interesting to note that acetic acid, a major inhibitory compound generated during many pretreatments, was not detected. Additionally, a phenolic content of 1.063 g/L was also present in the hydrolysate, indicating lignin breakdown during pretreatment. The concentration of inhibitory compounds basically depends on the type of biomass as well as the severity factor. Inhibitors can be classified on the basis of their chemical structure. Basically, inhibitors include furans, weak acids, and phenolics that are produced from the different fractions of the lignocellulosic biomass after the hydrothermal pretreatment. These inhibitors affect the rate of enzymatic hydrolysis as well as reduce glucose conversion during fermentation and affect the rate of ethanol formation at the end of biomass to ethanol formation [23,24].

The compositional analysis of hydrothermal pretreated pineapple leaves biomass revealed glucan content (% dry weight basis) of 33.55 ± 5.10%, xylan 19.7 ± 0.42%, and lignin 19.7 ± 2.16%. The total recovery of the solid fraction after the pretreatment was 83 g per 100 g of the raw biomass. The changes in the chemical composition of the pretreated biomass corresponded to significant lignin and xylan removal, as evident from the release of phenolics and xylose/arabinose in the pretreatment hydrolysate. However, cellulose breakdown into glucose during the pretreatment was undesirable and resulted in corresponding increases in the mass percentage of lignin and xylan. This could be targeted in future studies by employing more stringent pretreatment conditions with minimal loss of cellulosic sugars. Previous studies reported the minimum temperature required to break the short chains of cellulose to be 150 °C and that such milder hydrothermal pretreatment conditions can lead to hydrolysis of the amorphous regions of cellulose, resulting in release of glucose during the pretreatment [25,26]. The reported results were consistent with previous reports using similar or other lignocellulosic biomass, resulting in significant improvements in the cellulose content and improved enzymatic digestibility of the pretreated biomasses.

### 3.2. Enzymatic Hydrolysis of Pineapple Leaves Waste

In this study, the pretreated slurry and solid fraction obtained after liquid separation from the slurry after the hydrothermal pre-treatment were both subjected to enzymatic hydrolysis by commercial cellulase (Figure 1). During enzymatic hydrolysis of whole slurry by cellulase from *Trichoderma ressei* ATCC-26921, 4.8 g/L of glucose and 2.11 g/L of xylose were present initially, which increased to 13.72 g/L and 3.34 g/L, respectively, after 72 h of hydrolysis. Under similar conditions, hydrolysis of the solid fraction resulted in an increase in glucose and xylose concentrations from 0.62 and 0.33 g/L (0 h) to 18.38 g/L and 1.54 g/L (72 h), respectively. Total glucose yield of 367.7 mg/g and 274.4 mg/g was obtained after complete enzymatic hydrolysis of pretreated pineapple waste at a solid loading of 5% (dry wt.) in the solid and slurry’s fraction, respectively. Fractionation of the biomass to cellulose and hemicellulose after hydrothermal pretreatment resulted in initial higher glucose and xylose in the pretreatment slurry in comparison to that in the solid fraction due to extensive washing of the biomass. Thus, comparatively better glucose yield was obtained from hydrolysis of cellulose-rich solids compared to that from the whole slurry under similar hydrolysis conditions. Lower sugar yield from the slurry could be attributed to the presence of the inhibitory compounds in slurry generated during the pretreatment process, which could interfere with the enzyme performance. Previous studies have shown that xylooligomers [27,28] and phenolic compounds [29] act as inhibitors of the cellulase enzyme and their presence significantly reduces enzymatic hydrolysis efficiency. On the other hand, better hydrolysis of the solid fraction of pretreated biomass was attributed to the removal of most of the inhibitory compounds during the washing and solid-liquid separation steps prior to enzymatic hydrolysis of the biomass.

### 3.3. Fermentation of Pre-Treated Slurry and Cellulose-Rich Solid Fraction Hydrolysate

Hydrolysate obtained by hydrolysis of pineapple **leaves** waste using commercial cellulase was supplemented with nutrients and was further employed for ethanol production by Saccharomyces cerevisiae WLP300. Figure 2 shows ethanol production and sugar consumption during SHF of pretreated PL waste. The glucose concentrations in pretreated slurry and the cellulose-rich solid fraction were 13.72 g/L and 18.38 g/L, respectively, at the start of fermentation. The available glucose was utilized rapidly within 24 h of fermentation by yeast, with almost complete sugar consumption within 72 h by the yeast. The maximum ethanol concentrations of 6.4 g/L and 9.0 g/L, respectively, were obtained after the 72 h fermentation of enzymatic hydrolysates of whole slurry and solid fraction only. During fermentation, very small amounts of xylose present in the medium were also utilized. The calculated fermentation efficiencies (72 h) on basis of cellulose to ethanol conversion Equation (4) were 67.3% and 94.6% of the theoretical maximum for SHF of the whole slurry and solid fractions, respectively. The lower overall bioconversion efficiency in the case of whole slurry conversion could be mainly attributed to the lower cellulose hydrolysis by cellulases in presence of inhibitory compounds. However, on basis of available glucose to ethanol conversion, corresponding fermentation efficiencies were found to be 91.3 and 95.8%, respectively, indicating minimal effect of inhibitory compounds on yeast performance. Slightly lower ethanol efficiencies of 74.2 and 88.4% at 24 and 48 h, respectively, during fermentation of whole slurry enzymatic hydrolysate, as compared to that of 90.5 and 92.6% during fermentation of pretreated solids, could be attributed to the gradual adaptation of the yeast towards the fermentation inhibitors. Interestingly, near complete conversion (94.6% cellulose to ethanol conversion) of the cellulosic sugars from pretreated solids to bioethanol was observed in this study under SHF process. The results of bioethanol production from pineapple waste in the present study were better than previous reports of [30,31], who performed fermentation of corn stover biomass and olive tree biomass (OTB), respectively. The former research group reported the use of a high enzyme dose of 25 FPU/g for obtaining an ethanol titer of 32.2 g/L with 80% ethanol conversion efficiency, whereas a comparatively lower enzyme dose of 10 FPU/g was used in the present study with a correspondingly high fermentation efficiency. Martínez-Patiño and team reported an ethanol yield of 0.45 g/g sugar using a cellulase enzyme dose of 15 FPU/g supplemented with 15 units of beta-glucosidase, corresponding to fermentation efficiency of 88% and an ethanol yield of 0.144 g/g raw OTB [31]; the present study revealed maximum ethanol yields of 0.48 and 0.32 g/g, respectively, on the basis of conversion of available glucose and raw pineapple biomass to ethanol.

## 4. Conclusions

Overall, results of this study revealed the potential of pineapple waste biomass in the sustainable production of bioethanol and the role of inhibitors in recovery of sugars from pretreatment slurry when employed directly for bioethanol production. One of the comparative advantages of this study over previous studies has been the use of a more environmentally friendly hydrothermal pretreatment method without any acid or base catalyst. Moreover, this study employed whole hydrothermally pretreated pineapple leaves slurry for hydrolysis, indicating its feasibility for direct conversion without the need for washing step. Future studies may be directed to improving the cellulose recovery during pretreatment and detoxification of inhibitors as well as process optimization for achieving enhanced bioconversion of the whole biomass slurry as a substrate for bioethanol production.

## Figures and Tables

**Figure 1 bioengineering-09-00557-f001:**
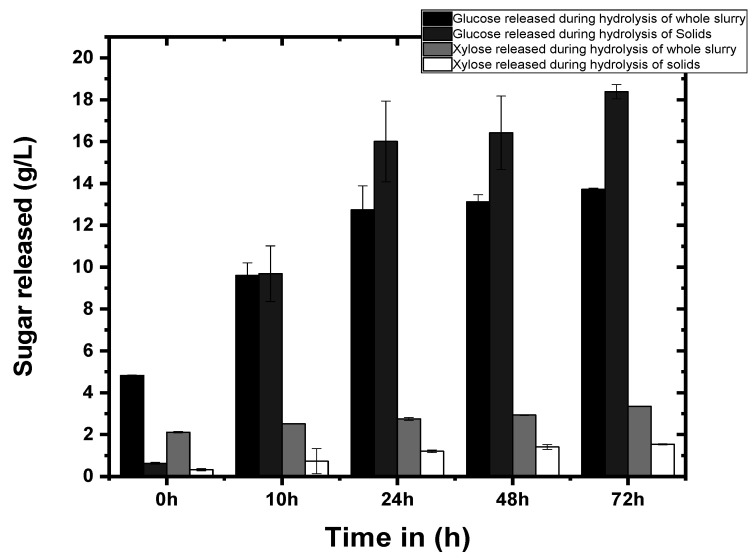
Enzymatic hydrolysis of whole slurry and solid fraction of hydrothermally pretreated pineapple leaves biomass.

**Figure 2 bioengineering-09-00557-f002:**
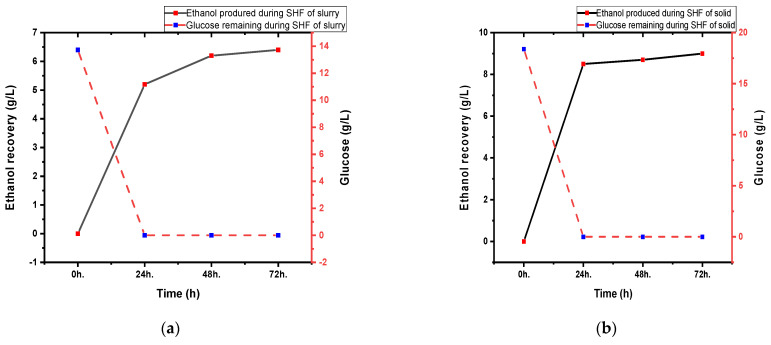
Ethanol production and sugar consumption profile during separate hydrolysis and fermentation at 30 °C of (**a**) whole slurry, and (**b**) solid fraction of hydrothermally pretreated pineapple leaves at 150 °C for 20 min. Ethanol production and glucose consumption are estimated by HPLC.

## Data Availability

Not applicable.

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
