# Peer review of "Valorization of Pineapple Leaves Waste for the Production of Bioethanol"

_bioengineering, 2022, doi:10.3390/bioengineering9100557_

Round 1

Reviewer 1 Report

R. R. Singahnia et al. submitted a paper on the valorization of pineapple leaves (not leaf !). In the present form, the present paper could not be accepted. The paper needs to be thoughtfully edited in order to be review.

1- citation and reference. Line 36, the authors cited a paper of Lee et al. 2019 for the annual pineapple production: “with an annual pineapple production of nearly 431,000 tons in 2019 (Lee et al., 2019).”

At the end of the manuscript, references are numbered. I suppose that reference 8, lines 325-326, corresponds to Lee et al., 2019 (lines 325-326). However, the title of the reference (lines 325-326) does not seem to be related to the annual pineapple production, i.e., “8. Lee, Y. C., Huang, S. L., & Liao, P. T. (2019). Land teleconnections of urban tourism: A case study of Taipei’s agricultural souvenir products. Landscape and Urban Planning, 191, 103616.”

2. The second paragraph (lines 52-89) could be shortened.

3. Hydrothermal pretreatment. In section 2.3, the solid residues are separated from the liquor. No information is given on the yield of this pretreatment. Furthermore, it is unclear which fraction the enzymatic hydrolysis (section 2.4) is performed. The authors used slurry (?) (line 135).

4. Calculation. How is determined the theoretical sugar yield and ethanol yield?

5. The authors justified the conditions of hydrothermal pretreatment by the work of Tânia Pinheiro et al. on brewer’s spent grain. However, brewer’s spent grain is different from pineapple leaves. The authors should cite and comment on the studies of Nashiruddin et al. in Industrial Crops and Products 152(2020) 112514 “Process parameter optimization of pretreated pineapple leaves fiber for enhancement of sugar recovery” as well as SaksitImman et al. Energy Reports 7(2021) 6945-6954 “Optimization of sugar recovery from pineapple leaves by acid-catalyzed liquid hot water pretreatment for bioethanol production”. Why was 20 min at 150°C chosen?

Author Response

Reviewer’s Comments

Authors Response

R1

R. R. Singahnia et al. submitted a paper on the valorization of pineapple leaves (not leaf !). In the present form, the present paper could not be accepted. The paper needs to be thoughtfully edited in order to be review.

The title of the manuscript has now been revised as per reviewer’s suggestion. (L1-3)

1- citation and reference. Line 36, the authors cited a paper of Lee et al. 2019 for the annual pineapple production: “with an annual pineapple production of nearly 431,000 tons in 2019 (Lee et al., 2019).” At the end of the manuscript, references are numbered. I suppose that reference 8, lines 325-326, corresponds to Lee et al., 2019 (lines 325-326). However, the title of the reference (lines 325-326) does not seem to be related to the annual pineapple production, i.e., “8. Lee, Y. C., Huang, S. L., & Liao, P. T. (2019). Land teleconnections of urban tourism: A case study of Taipei’s agricultural souvenir products. Landscape and Urban Planning, 191, 103616.”

Thank you so much for pointing out this critical issue.

The earlier cited paper by Lee et al 2018 has been replaced by the correct reference/citation and the data on pineapple annual production has been accordingly updated

2. The second paragraph (lines 52-89) could be shortened.

The paragraph has been shortened by deleting the repeated and redundant information.

3. Hydrothermal pretreatment. In section 2.3, the solid residues are separated from the liquor. No information is given on the yield of this pretreatment. Furthermore, it is unclear which fraction the enzymatic hydrolysis (section 2.4) is performed. The authors used slurry (?) (line 135).

The information on the pretreatment yield has been provided in the results. The authors have used both slurry as well as washed biomass for hydrolysis. The referred line 135 pertains to the compositional analysis of the pretreated biomass, for which of course the pretreated solids could only be used.

4. Calculation. How is determined the theoretical sugar yield and ethanol yield?

Theoretical yield of saccharification was based upon the theoretically possible maximum sugar release from the biomass cellulose fraction (0.33).

Theoretical ethanol yield was based upon the sugar to ethanol conversion factor of 0.51

5. The authors justified the conditions of hydrothermal pretreatment by the work of Tânia Pinheiro et al. on brewer’s spent grain. However, brewer’s spent grain is different from pineapple leaves.

The authors should cite and comment on the studies of:

Nashiruddin et al. in Industrial Crops and Products 152(2020) 112514 “Process parameter optimization of pretreated pineapple leaves fiber for enhancement of sugar recovery” as well as

SaksitImman et al. Energy Reports 7(2021) 6945-6954 “Optimization of sugar recovery from pineapple leaves by acid-catalyzed liquid hot water pretreatment for bioethanol production”.

Why was 20 min at 150°C chosen?

Nashiruddin et al. have used temperature of 80 to 100 °C along with the chemical catalyst (Dilute sulfuric acid / Sodium hydroxide). Similarly, SaksitImman et al. employed the catalyst (sulfuric acid at concentration of 0.61 M) and performed the LHW treatment at 120-160 C for 20-50 min. However, the present study was focused on carrying out the LHW pretreatment of pineapple leaves without employing any catalyst and needed slightly higher temperature than when using the catalyst. We have now cited this paper by Saksit Imman et al. in the paper as this condition chosen were closer to that used in present study.

Moreover, preliminary experiments conducted at 150, 170 and 190 °C (data not shown) indicated that use of the former temperature condition worked better and resulted in more sugar release in the hydrolysate along with less inhibitors than at 170 or 190 °C.

Therefore, based on preliminary lab experiments and the literature review, we selected the chosen conditions for pretreatment of PL.

Reviewer 2 Report

The manuscript is well prepared but there are many studies available on the production of bioethanol from pineapple leaf waste such as https://doi.org/10.1016/j.jclepro.2017.07.179; https://doi.org/10.1016/j.biombioe.2020.105675; https://doi.org/10.1016/j.jclepro.2017.10.284; https://doi.org/10.1016/j.egyr.2021.10.076 among many others. Authors should clearly describe the novelty of this study. I kindly recommend the resubmission of manuscript after major revisions focusing on the novelty of this study.

Author Response

We are grateful for Editor and reviewers who devoted time to read our manuscript and gave insightful comments which helped us to improve our overall manuscript.

Response to Reviewer 2 comments

Reviewer Comments

Author’s response

The manuscript is well prepared but there are many studies available on the production of bioethanol from pineapple leaf waste such as https://doi.org/10.1016/j.jclepro.2017.07.179; https://doi.org/10.1016/j.biombioe.2020.105675; https://doi.org/10.1016/j.jclepro.2017.10.284; https://doi.org/10.1016/j.egyr.2021.10.076 among many others. Authors should clearly describe the novelty of this study. I kindly recommend the resubmission of manuscript after major revisions focusing on the novelty of this study.

These all studies are interesting and insightful, however,the present study was based on greener pretreatment (hydrothermal pretreatment without any acid and base catalyst) method. Moreover, this study attempted to use whole hydrothermally pretreated pineapple leaves slurry for hydrolysis indicating its feasibility for direct conversion without the need for washing step.  However, more focused studies are needed in this direction in future for improving the yield of enzymatic conversion of whole pretreatment slurry.

As per the suggestion we have included novelty statement in the last paragraph of the introduction.

Reviewer 3 Report

The article is interesting, but there are a few things that need to be clarified: Fig. 2 must also show the working conditions (temperature, pressure, etc.). I suggest the authors to present synthetically in the conclusions the comparative advantage and the novelty of the research Authors are also requested to review the papers:

COMBUSTION OF BIOGAS OBTAINED BY ANAEROBIC FERMENTATION OF ANIMAL PROTEINS, in INNOVATIVE RENEWABLE WASTE CONVERSION TECHNOLOGIES, ISBN 978-3-030-81430-4, ISBN (eBook): 978-3-030-81431-1, doi.org/10.1007/978-3-030-81431-1_6, Publisher: Springer Nature

EXPERIMENTAL RESEARCH ON COMBUSTION OF BIOGAS OBTAINED THROUGH ANAEROBIC FERMENTATION OF TANNERIES WASTES, UNIVERSITY POLITEHNICA OF BUCHAREST SCIENTIFIC BULLETIN SERIES B-CHEMISTRY AND MATERIALS SCIENCE, ISSN: 1454-2331, Volume: 80  Issue: 3  Pages: 105-116, Article Number: 3469, 2018, Accession Number:  WOS:000440890800010

Author Response

We are grateful for Editor and reviewers who devoted time to read our manuscript and gave insightful comments which helped us to improve our overall manuscript.

Response to Reviewer 3 comments

Reviewers comments

Authors Response

The article is interesting, but there are a few things that need to be clarified: Fig. 2 must also show the working conditions (temperature, pressure, etc.).

I suggest the authors to present synthetically in the conclusions the comparative advantage and the novelty of the research

As suggested, the information on temperature condition has been added in the Fig. 2 legend. However, since the figure represents the data on biomass hydrolysis and the pressure is not applied during hydrolysis, therefore, the same has not been mentioned.

As far as the novelty is concerned, as indicated in response to preceding comment above, the present study was based on greener pretreatment (hydrothermal pretreatment without any acid and base catalyst) method. Moreover, this study attempted to use whole hydrothermally pretreated pineapple leaves slurry for hydrolysis indicating its feasibility for direct conversion without the need for washing step.  However, more focused studies are needed in this direction in future for improving the yield of enzymatic conversion of whole pretreatment slurry.

Authors are also requested to review the papers:

COMBUSTION OF BIOGAS OBTAINED BY ANAEROBIC FERMENTATION OF ANIMAL PROTEINS, in INNOVATIVE RENEWABLE WASTE CONVERSION TECHNOLOGIES, ISBN 978-3-030-81430-4, ISBN (eBook): 978-3-030-81431-1, doi.org/10.1007/978-3-030-81431-1_6, Publisher: Springer Nature

EXPERIMENTAL RESEARCH ON COMBUSTION OF BIOGAS OBTAINED THROUGH ANAEROBIC FERMENTATION OF TANNERIES WASTES, UNIVERSITY POLITEHNICA OF BUCHAREST SCIENTIFIC BULLETIN SERIES B-CHEMISTRY AND MATERIALS SCIENCE, ISSN: 1454-2331, Volume: 80  Issue: 3  Pages: 105-116, Article Number: 3469, 2018, Accession Number:  WOS:000440890800010

These articles are very interesting which will be considered in our upcoming review article on ‘biomethane advances’.

In the present study we did not find any scope to cite these articles as our present manuscript is solely based on bioethanol production.

Round 2

Reviewer 1 Report

The authors modified their manuscript according to the reviewer's comments. However, I still have difficulty understanding their results and comparisons.

After the thermal pretreatment of grounded pineapple leaves (water, 150°C, 20 min), the solid residues were separated from the liquor by centrifugation and washing.

1/ What amount of solid residues per g of grounded pineapple leaves? We have this information on lines 226-228. This information should also be added in section 2.3. (see also point 5/).

2/It is claimed (lines 141-143) that the composition of solid and liquor will be determined and compared in section "analytical methods". However, in section 2.6, I do not see any solid and liquor characterization.

3/In section 3.1, lines 215-217, we have an HPLC analysis of the liquors "HPLC analysis revealed glucose, xylose and arabinose contents of 15.95, 20.7 and 2.0 g/L, respectively,". The concentration of sugars is useless since the solid was washed with an unknown amount of water (lines 139-141, "The recovered solids were washed exhaustively with water until neutral pH and squeezed before preservation at -20 °C for further use.”)

4/ Lines 198-200, one can read: "The compositional analysis of untreated pineapple leaves biomass determined on a dry weight basis revealed glucan content of 56.90 ± 2.10 %, xylan 10.88 ± 0.35 % and lignin 14.2± 0.42%." How is this analysis performed? What are the units?

Lines 225-226, one can read: "The compositional analysis of hydrothermal pretreated pineapple leaves biomass (solid ?)revealed glucan content of 33.55 ± 5.10%, xylan 19.7 ± 0.42% and lignin 19.7 ± 2.16%." How is this analysis performed? What are the units?

5/ A table that compares the initial composition of untreated pineapple leaves, the solid, and the liquor should be added to understand the mass balance.

6/The authors should name throughout the manuscript the liquor, liquor (line 145), and not liquor then slurry (line 148).

7/ In section 3.2, enzymatic hydrolysis of the solid and the liquor is performed. I do not understand the legend in figure 1. What does "Glucose concentration in slurry" mean and "Glucose concentration in solid"? It is not the concentration of glucose in the solid but the amount of glucose released.

Again we have no information on the amount of solid and liquor used in the enzymatic hydrolysis. Thus a concentration of glucose in the solution is useless.

8/ The authors claim that the better hydrolysis of the solid (see point 5/) could be explained by removing "inhibitory compounds". Could the authors give the readers more information on these inhibitory compounds?

In conclusion, the bibliography has been improved,a but the analytical section and discussion need to be revised before publication.

Author Response

We thank the reviewers and editors for insightful comments, which has enabled us to improve our manuscript. We have addressed each query with utmost sincerity. We believe that the manuscript will reach to a satisfactory level to be accepted in Bioengineering.

Reviewer’s Comments

Comment: What amount of solid residues per g of grounded pineapple leaves? We have this information on lines 226-228. This information should also be added in section 2.3. (see also point 5/).

Response: The information on amount of solid residues per g of grounded pineapple leaves has been added in section 2.3, Line 131

Comment: It is claimed (lines 141-143) that the composition of solid and liquor will be determined and compared in section "analytical methods". However, in section 2.6, I do not see any solid and liquor characterization.

Response: The author has checked line 141-143. These lines refer to the characterization of pretreated and untreated biomass and not solid liquid characterization as referred in the comment by the reviewer. The method for biomass characterization has been described in section 2.6.1. Characterization of liquor has been described in section 2.6.2 and the text has been revised to better indicate the characterization of liquor obtained after pretreatment. Line 166-168. The composition analysis is given in line 215-221.

Comment: In section 3.1, lines 215-217, we have an HPLC analysis of the liquors "HPLC analysis revealed glucose, xylose and arabinose contents of 15.95, 20.7 and 2.0 g/L, respectively,". The concentration of sugars is useless since the solid was washed with an unknown amount of water (lines 139-141, "The recovered solids were washed exhaustively with water until neutral pH and squeezed before preservation at -20 °C for further use.”)

Response: The authors would like to clarify to the reviewer that the solid obtained after the pretreatment was washed until neutral pH to remove the inhibitors from biomass for its efficient enzymatic hydrolysis. The HPLC analysis of the liquor obtained after the separation of solids from the pretreatment slurry was performed.

Comment: Lines 198-200, one can read: "The compositional analysis of untreated pineapple leaves biomass determined on a dry weight basis revealed glucan content of 56.90 ± 2.10 %, xylan 10.88 ± 0.35 % and lignin 14.2± 0.42%." How is this analysis performed? What are the units?

Response: The composition analysis of treated and untreated pineapple leaves was determined as per the protocol by Sluiter et al., (2008), on basis of (% dry weight) and is given in Line 215-221.

Comment: Lines 225-226, one can read: "The compositional analysis of hydrothermally pretreated pineapple leaves biomass (solid?) revealed glucan content of 33.55 ± 5.10%, xylan 19.7 ± 0.42% and lignin 19.7 ± 2.16%." How is this analysis performed? What are the units?

Response: The compositional analysis of untreated and hydrothermal pre-treated pineapple leaves was performed as per the protocol of Sluiter et al., (2008). The units are already mentioned “on % dry weigh(= g/100g recovered solids)”

Comment: A table that compares the initial composition of untreated pineapple leaves, the solid, and the liquor should be added to understand the mass balance.

Response: The data on the composition of untreated and pretreated solids as well as the sugars and inhibitors present in the liquor is already mentioned in the text in lines 198-200, 225-226, and 216-224 respectively therefore the addition of the table as per the reviewer’s suggestion will be a repetition of the data.

Comment: The authors should name throughout the manuscript the liquor, liquor (line 145), and not liquor then slurry (line 148).

Response: The authors would like to clarify that liquor and slurry are two separate things. The liquor line 145 refers to the liquid obtained after the removal of the solids from the process of centrifugation from the pretreatment slurry. The slurry (line 148) was used (which contains both the pretreated solids and the liquor) was used for the enzymatic hydrolysis experiments, not the liquor. The authors have revised lines 145 and 148 to remove the confusion between liquor and slurry.

Comment: In section 3.2, enzymatic hydrolysis of the solid and the liquor is performed. I do not understand the legend in figure 1. What does "Glucose concentration in slurry" mean and "Glucose concentration in solid"? It is not the concentration of glucose in the solid but the amount of glucose released

Again, we have no information on the amount of solid and liquor used in enzymatic hydrolysis. Thus, a concentration of glucose in the solution is useless.

Response: We agree that glucose concentration in solid does not mean its content but it means glucose released from the solid fraction. The authors agree with the reviewer that the legend in figure 1 was somewhat confusing. Therefore, the legend has been corrected in the revised manuscript to indicate the amount of glucose released during the enzymatic hydrolysis of the solid and the slurry. Accordingly, Fig2 and the text of its legend have also been modified in the revised manuscript.

As far as the amount of solid and slurry used in enzymatic hydrolysis has been already mentioned in line 255 (5% dry-weight solid loading).

Comment: The authors claim that the better hydrolysis of the solid (see point 5/) could be explained by removing "inhibitory compounds". Could the authors give the readers more information on these inhibitory compounds?

In conclusion, the bibliography has been improved, but the analytical section and discussion need to be revised before publication.

Response: More information on different inhibitory compounds has been added in the text as per the reviewer’s comments.

We have tried to improve discussions as per the suggestions, see Line 215-225.

Analytical section: Line 166-168

Conclusions: Line 314-319

Reviewer 2 Report

The manuscript has been improved and can be accepted.

Author Response

Thank you so much. Authors are grateful for reviewers hard work to check our modifications.

Reviewer 3 Report

MAJOR REVISION

Author Response

Dear Reviewer we have tried to improve all the sections as per your kind report. We hope you may find this revised version of our manuscript to your satisfactory level.